Factors and associations for physical activity in severely obese adults during a two-year lifestyle intervention

Jepsen Randi 1 2 randi.jepsen@hisf.no
Aadland Eivind 1
Robertson Lesley 3
Kristiansen Merete 4
Andersen John Roger 1 5
Natvig Gerd Karin 2
1 Faculty of Health Studies, Sogn og Fjordane University College , Førde , Norway
2 Department of Global Public Health and Primary Care, University of Bergen , Bergen , Norway
3 Red Cross Haugland Rehabilitation Centre , Flekke , Norway
4 Faculty of Teacher Education and Sport, Sogn og Fjordane University College , Sogndal , Norway
5 Førde Health Trust , Førde , Norway
Ehrlich-Jones Linda
Electronic publication date: 2014 Aug 7
Publication date: 2014
Volume: 2
Electronic Location ID: e505
Received 2014 May 19; Accepted 2014 Jul 18
Copyright: © 2014 Jepsen et al.
Copyright year: 2014
Copyright holder: Jepsen et al.
License: This is an open access article distributed under the terms of the Creative Commons Attribution License, which permits unrestricted use, distribution, reproduction and adaptation in any medium and for any purpose provided that it is properly attributed. For attribution, the original author(s), title, publication source (PeerJ) and either DOI or URL of the article must be cited.
License URL: https://creativecommons.org/licenses/by/4.0/

Keywords: Severe obesity, Physical activity, Lifestyle intervention, Self-identity, Self-efficacy, Perceived behavioural control, Accelerometer

Funding: Faculty of Health Studies, Sogn og Fjordane University College The study was financially supported by the Faculty of Health Studies, Sogn og Fjordane University College, Norway. The funders had no role in study design, data collection and analysis, decision to publish, or preparation of the manuscript.

==============================
Objective. This study of severely obese adults participating in a two-year lifestyle intervention investigates associations between the independent variables: change in self-efficacy for physical activity (PA) in the face of psychological barriers, perceived behavioural control over PA, and PA self-identity and the dependent variable of change in objectively assessed PA. The intervention comprised four residential periods in a rehabilitation centre and combined diet, physical activity, and cognitive behavioural therapy.

Materials and Methods. Forty-nine severely obese adults (37 women, mean body mass index 42.1 kg/m2) were included in the study. Assessment was done four times using questionnaires and an accelerometer. A linear mixed model based on restricted maximum likelihood was used in analyses for change over time. Associations were studied using linear regression analyses. Age, gender, and change in body mass index were used as control variables.

Results. In the adjusted analyses, change in perceived behavioural control over PA was associated with change in PA (Stand. coeff. = 0.32, p = .005). Change in PA was not associated with either change in self-efficacy over PA in the face of psychological barriers (Stand. coeff. = 0.13, p = .259) or PA self-identity (Stand. coeff. = −0.07, p = .538).

Conclusion. Perceived behavioural control may be a valid target to increase and maintain PA in severely obese adults participating in lifestyle interventions. More research is needed to investigate the process of behaviour change in this population.

Introduction

A web of political, societal, and environmental factors contributes to the growing prevalence of overweight and obesity in Western countries (Swinburn et al., 2011). Alarmingly, the group classified as severely obese has increased the most (Sturm, 2007; Midthjell et al., 2013). In obesogenic societies (Swinburn et al., 2011), the health services have responsibility to ameliorate the ailing health, functioning, and quality of life experienced by severely obese individuals (van Nunen et al., 2007). Thus, various obesity surgeries and lifestyle interventions have been developed. Obesity surgery brings about physiological and functional changes, enforcing altered eating behaviours and thus leading to weight loss (Kissler & Settmacher, 2013). By contrast, lifestyle interventions work exclusively through individual modifications and self-management of health-related behaviour (Kirk et al., 2012). A weight loss of 5–10% is regarded as sufficient to gain health effects and reduce the risk of obesity-related comorbidities (Tsigos et al., 2008; Dalle Grave, Calugi & El Ghoch, 2013). The most extensive weight losses are obtained through obesity surgery (Karlsen et al., 2013), whereas subjects attending lifestyle interventions seem to develop more favourable dietary patterns (Johnson et al., 2013). Both intervention types improve health-related quality of life (Karlsen et al., 2013).

In combination with diet modification, physical activity (PA) constitutes a core component of many lifestyle interventions for severely obese adults (Kirk et al., 2012; Dalle Grave, Calugi & El Ghoch, 2013). Studies have shown that PA impacts on weight loss and its maintenance (Catenacci & Wyatt, 2007; Butryn, Webb & Wadden, 2011), improves body composition (Lee et al., 2005; Kay & Fiatarone Singh, 2006; Goodpaster et al., 2010), reduces risk of cardio-metabolic comorbidities (Fogelholm, 2010; Goodpaster et al., 2010), and is positively associated with quality of life (Bond et al., 2006; Lerdal et al., 2011; Jepsen et al., 2013) in severely obese subjects. Thus, given the chronic nature of severe obesity, adherence to PA is important (Tsigos et al., 2008), but unfortunately PA decreases with increasing body mass index (BMI) (Tudor-Locke et al., 2010; Hansen et al., 2013). Severely obese subjects face many barriers to PA, such as the excess body weight itself (Wiklund, Olsén & Willén, 2011; Christiansen, Borge & Fagermoen, 2012) and exposure in public (Wiklund, Olsén & Willén, 2011). A persistent increase of PA seems to be difficult (Borg et al., 2002; Tate et al., 2007). Thus, lifestyle interventions should target and strengthen patients’ resources for PA through provision of knowledge and skills and reinforcement of psychological factors that are likely to influence PA. Hence, self-efficacy for PA, perceived behavioural control over PA, and PA self-identity have been proposed as targets for PA interventions (Hagger, Chatzisarantis & Biddle, 2002; Jackson, Smith & Conner, 2003; Lorentzen, Ommundsen & Holme, 2007; Hansen et al., 2014).

According to Bandura (1997), self-efficacy covers “a belief about what one can do under different sets of conditions with whatever skills one possesses” (p. 37). Self-efficacy is dynamic and modifiable (Bandura, 1997), and in relation to PA, it includes the capability of adoption and maintenance of PA in the face of psychological barriers such as feeling depressed, worried, angry, or stressed (Lorentzen, Ommundsen & Holme, 2007). Self-efficacy for PA in the face of psychological barriers has shown positive associations with change in PA in community samples (Lorentzen, Ommundsen & Holme, 2007). The related concept of perceived behavioural control refers to a person’s “belief as to how easy or difficult performance of the behaviour is likely to be” (Ajzen & Madden, 1986, p. 457). Perceived behavioural control predicts PA behaviour (Hagger, Chatzisarantis & Biddle, 2002) and plays a role in PA behaviour change in community samples (Lorentzen, Ommundsen & Holme, 2007). Identity is a third factor influencing human behaviour. Hence, there is a reciprocal reinforcing relationship between a behaviour-specific identity and repetition of that behaviour. Furthermore, identity is a product of interaction with others (Charng, Piliavin & Callero, 1988). PA identity, i.e., “identifying oneself as a physically active person” (Lorentzen, Ommundsen & Holme, 2007, p. 95), has shown positive associations with self-reported PA (Jackson, Smith & Conner, 2003) and change in PA (Lorentzen, Ommundsen & Holme, 2007) in community samples.

Common outcome variables in research on lifestyle interventions for severely obese adults are body weight and risk factors for medical comorbidities (Anderson, Conley & Nicholas, 2007; Goodpaster et al., 2010; Danielsen et al., 2013; Karlsen, Sohagen & Hjelmesaeth, 2013). Because the pathway for these outcomes is behaviour change, it is pivotal to understand factors involved in the behaviour change process. However, to our knowledge, no studies have investigated associations in patterns of change between psychological factors and objectively assessed PA in severely obese adults during a lifestyle intervention. Thus, the aim of this study was to investigate associations between the independent variables: change (Δ) in self-efficacy, Δ perceived behavioural control, and Δ self-identity and the dependent variable of Δ PA. The study hypothesis was that there would be positive associations in the patterns of change between self-efficacy for PA in the face of psychological barriers, perceived behavioural control over PA, and PA identity and objectively assessed PA in severely obese adults participating in a two-year lifestyle intervention.

Materials and Methods

Participants and setting

The Haugland Obesity Study has a two-year prospective design. We assessed severely obese patients before, during, and between residential periods in Red Cross Haugland Rehabilitation Centre (RCHRC) in Norway, where they participated in a two-year lifestyle intervention. The programme was funded by the public health services. For those with jobs, the social welfare system paid sick leave benefits during the residential periods. Referral to RCHRC was done by general practitioners. Referred patients were called in to a two-week stay, wherein motivation for change and ability to function in a group were assessed and participation in the programme was decided. Due to limited capacity of the centre, the time from referral to this assessment stay was up to two years. Fifty-three patients, divided in four groups, started the actual intervention (the time point which constituted the baseline of the present study) and were all offered inclusion in the present study. After the intake of these four groups, the public health services reduced the funding to RCHRC and cut the intervention for new patients to a one-year programme with shorter residential stays. Thus, we had to discontinue inclusion of participants.

Inclusion criteria for the intervention were age between 18 and 60 years, and BMI ≥ 40 kg/m2 with or without comorbidities, or ≥ 35 kg/m2 with comorbidities, such as type 2 diabetes, treatment-resistant hypertension, and sleep apnoea (Norwegian Directorate of Health, 2009). Exclusion criteria were: referral to, or, previous obesity surgery; pregnancy; severe cardiovascular disease; alcohol or substance abuse; and mental illness or physical impairment that would prevent adherence to the intervention.

Intervention

The lifestyle intervention was a multi-component programme developed by the health professionals at RCHRC, built on recommendations for best practice (Shaw et al., 2005; Kirk et al., 2012; Dalle Grave, Calugi & El Ghoch, 2013; Olander et al., 2013) and prior experience. The objective was to strengthen favourable PA and diet-related behaviour. The intervention covered 15 weeks over two years with four residential periods of six, three, three, and three weeks’ duration. A team of physicians, nurses, physiotherapists, exercise therapist, and dietician–many with training in cognitive behavioural therapy (CBT) (Shaw et al., 2005)–delivered mandatory practical and theoretical sessions on PA, diet, and CBT. The PA consisted of supervised and un-supervised out- and indoor activities, in groups and individually. Brisk walking, swimming, strength training, ball games, and aerobics were the predominant activities during the residential periods, bringing on moderate to high intensities of PA. The scheduled PA lasted 20–60 min per session, in total nine to eleven hours per week. In addition, the patients were encouraged to carry out PA on their own initiative. Taking preferences, limitations, and sustainability into account, each patient developed a plan for PA for home periods and discussed it with staff. Patients were recommended to combine endurance and strength training and undertake at least 60 daily minutes of PA, which could be divided in intervals of down to ten minutes (Becker et al., 2004). Thus, no standard exercise programme was performed in the home periods. The main goal regarding diet was adaptation to a sustainable, healthy diet and a favourable eating-pattern. The meal plan at RCHRC was based on the Nordic Nutrition Recommendations (Becker et al., 2004) with three low-fat, high-fibre, and energy-reduced meals and two to three snacks per day. Most of the meals were provided by the canteen. However, the patients prepared some of their meals in supervised groups, and they ate together. They were advised to follow the same dietary principles at home. Thus, severe energy-restriction was not applied. In total, eleven group sessions of CBT, led by members of the health care team, took place, five during the six-week residential period and two during each of the subsequent three three-week periods. Before the end of each session, home work was given, and the following session started with a review of that. In sessions 1–5, during the first stay, the methods and instruments of CBT were introduced and related to change in PA and eating. When the patients came back for the subsequent periods, experiences from the home periods were discussed in sessions 6–11 within the framework of CBT. For eight patients who expressed a need, individual CBT was also provided.

Planning, barrier identification, and relapse prevention and management were practiced to strengthen self-management of PA and eating (Olander et al., 2013). The group-based activities aimed at stimulating peer support (Olander et al., 2013). Self-monitoring in home periods was promoted using PA diaries (Olander et al., 2013) in which patients reported on PA and a few added information on diet and success stories. All patients sent their diaries to RCHRC every month. There was no other structured follow-up between the residential periods. Patients were encouraged to contact their general practitioner if they needed more support and relatives were not involved in the intervention.

Measures and procedures

Data were collected four times between February 2010 and October 2012 (Fig. 1). The first collection, baseline, was done prior to the start of the intervention, the second at the end of the first residential stay and the third before the third stay, one year from baseline. The final data collection took place before the fourth and last stay, two years from baseline. Socio-demographic data in this study are baseline data.

Figure 1 Flow chart for the two-year follow-up study of severely obese adults in a lifestyle intervention.

PA, physical activity.

Psychological factors

The psychological factors were assessed using self-reported questionnaires. Self-efficacy for PA was assessed using a five-item measure. The participants indicated the extent to which they were confident in their ability to perform planned PA in the face of psychological barriers (i.e., feeling tired, depressed, anxious, angry, and stressed) on a seven-point scale from 1 (“not at all confident”) to 7 (“very confident”). The scale is a shortened and moderated version of the original instrument developed by Bandura (2001). The version used in this study has demonstrated excellent internal consistency with a Cronbach alpha coefficient of 0.91 (Hansen et al., 2014). Another five-item measure was used to assess perceived behavioural control. The participants rated their agreement with three positive (e.g., “I have total control over being regularly physically active”) and two negative statements (e.g., “Being regularly physically active is difficult for me”) on a seven-point scale from 1 (“totally agree”) to 7 (“don’t agree at all”). The scale is an extended and moderated version of the original instrument developed by Norman & Smith (1995). The version used in this study has demonstrated acceptable internal consistency with a Cronbach alpha coefficient of 0.67 (Hansen et al., 2014). PA self-identity was assessed using a three-item measure. The participants indicated the extent to which they agreed with statements such as “Being physically active is a big part of who I am” on a five-point scale from 1 (“fits poorly”) to 5 (“fits well”). The scale is a shortened and moderated version of the original instrument developed by Anderson & Cychosz (1994). The version used in this study has demonstrated excellent internal consistency with a Cronbach alpha coefficient of 0.91 (Hansen et al., 2014). All three instruments have shown positive cross-sectional associations with objectively assessed PA in adults (Hansen et al., 2014).

Physical activity

PA was measured using the accelerometer Actigraph GTI M (Actigraph, Fort Walton Beach, FL, USA), which is a hip-worn electronic movement sensor that converts acceleration into the arbitrary unit “counts”. The counts increase with the magnitude of the work rate for walking. The participants were instructed to wear the accelerometer on the right hip for seven consecutive days, except while sleeping or during water activities. The second assessment was completed towards the end of the first residential stay whereas the others took place in home periods (Fig. 1). The data were analysed with the Actigraph software ActiLife v. 5.3. A wear-time of ≥ ten hours per day for ≥ four days was the criterion for a valid measure. Periods of ≥60 consecutive minutes without counts were defined as non-wear-time, allowing for up to two minutes of counts greater than zero within these 60 min (Trost, McIver & Pate, 2005; Sirard et al., 2011). The counts were summed and averaged over the total wear-time to indicate the overall PA in counts per minute (CPM). The accelerometer has been found to be valid in severely obese adults (Aadland & Anderssen, 2012) and accelerometer-assessed PA has superior validity compared to self-reported data (Prince et al., 2008).

Socio-demographic information and anthropometry

Socio-demographic information was obtained from questionnaires. Height was measured without shoes to the nearest 0.5 cm with a wall mounted stadiometer (SECA, Germany). Weight was measured on a bioelectrical impedance analysis device (BC 420S MA, Tanita Corp, Tokyo, Japan) and reported to the nearest 0.1 kg.

Ethics

Written informed consent was obtained from all participants prior to the data collection, in accordance with the Helsinki Declaration. Ethical approval was obtained from the Regional Committee for Medical and Health Research Ethics for South-East Norway (registration number 2010/159).

Statistical analysis

Before calculating the mean values for the psychological factors, the three positively worded items for perceived behavioural control were reversed. Thus, higher mean values indicated stronger self-efficacy, perceived control, and identity. Cronbach alpha was used to determine the internal consistency of the instruments.

Data on civil status were dichotomized into “married/cohabiting” vs “single/divorced”, educational level into “<15 years of education” vs. “≥15 years of education” (i.e., college/university), and employment into “not working” (i.e., being unemployed or receiving pensions or benefits) vs. “working”. BMI was calculated as weight in kilograms divided by the square of the height in meters.

A linear mixed model based on restricted maximum likelihood estimation with random intercept for subjects was used in all analyses for change over time (Twisk, 2003), using least significant difference from baseline. Effect size (ES) for change was calculated by subtracting the two-year score from the baseline score, divided by the standard deviation (SD) at baseline. ES were judged against the standard criteria proposed by Cohen: Small change (0.2 to <0.5), moderate change (0.5 to <0.8), and large change (≥0.8) (Ellis, 2011).

The associations between the independent variables: Δ self-efficacy for PA in the face of psychological barriers, Δ perceived behavioural control over PA, and Δ PA self-identity and the dependent variable of Δ PA were analysed using linear regression, applying delta scores between time points (Δy1 = y1−y0; Δx1 = x1−x0; Δy2 = y2−y1, etc.) (Twisk, 2003). For the independent and dependent variables and BMI, the differences between baseline and week six (Δ1), between week six and year one (Δ2), and between year one and year two (Δ3) were used. The linear mixed model was omitted because the interpretation of the regression coefficients in such a model is difficult, due to mixing of longitudinal (with-in subject) changes and the cross-sectional (between-subject) differences (Twisk, 2003). Age, gender, and Δ BMI served as covariates in the multiple regression analyses. A total of N = 71 observations was included in the regression analyses. Residuals were normally distributed in all models.

Baseline subject characteristics are presented as percentages for categorical data and mean values (SD) for continuous variables. The estimates, obtained from the linear mixed model, for the psychological factors, PA, and BMI are presented as means with 95% confidence intervals (CI) for the four assessment points. We performed a drop-out analysis with the chi-squared test for difference in gender and the independent samples t-test for differences in other variables.

The statistical analyses were done using SPSS v. 20.0 (SPSS Inc., Chicago, USA). A two-sided p-value ≤ 0.05 indicated statistical significance.

Results

Forty-nine patients (37 women, 75.5%) consented to participate in the study. Baseline characteristics are presented in Table 1. Other details of the participants have been presented previously (Jepsen et al., 2013; Aadland et al., 2014).

Table 1 Characteristics of the study sample at baseline, N = 49.

Age, mean (SD)	43.6 (9.4)	
Gender, n (%)		
Women	37 (75.5)	
Socio-demographic status, n (%)		
Married/cohabiting	30 (61.2)	
Having children	27 (55.1)	
Formal education ≥ 15 years	22 (44.9)	
Employed	41 (83.7)	
Anthropometrics, mean (SD)		
Body mass index, kg/m2	42.1 (6.0)	
Notes.

SD Standard deviation

Drop-outs and available data for all time points are displayed in Fig. 1. At year two, twenty-two participants (44.9%, 16 women and six men) were lost to follow-up. Reasons for dropping out of the intervention included pregnancy, referral to obesity surgery, having reached personal weight goal, health problems, or obligations that interfered with the intervention. Six participants dropped out for unknown reasons and five withdrew from the study due to problems with the study protocol (repeated blood tests and assessments of maximal oxygen consumption which were included in the Haugland Obesity Study). The participants lost to follow-up did not differ from those who completed the study with regards to gender, age, BMI, PA, or psychological factors at baseline, or initial changes (during the first six weeks) in BMI, PA, or psychological factors. Missing data for psychological factors resulted from participants being absent when the questionnaires were administered at RKHRC. Furthermore, some of the accelerometer-obtained data failed to fulfil the validity requirements.

The internal consistency of the measures of self-efficacy, perceived behavioural control, and self-identity, calculated at baseline, were acceptable to excellent using Cronbach alpha coefficients of 0.92, 0.67, and 0.93, respectively.

Table 2 shows that PA increased significantly from baseline to the end of the first residential period and remained increased at the one-year follow-up. However, after two years the increase in PA was not maintained (ES = 0.24). All three psychological factors were significantly strengthened at the end of the first residential period (Table 2). However, self-efficacy for PA in the face of psychological barriers decreased thereafter and at one year the improvement had vanished (ES = 0.14). In contrast, perceived behavioural control over PA (ES = 0.51) and PA self-identity (ES = 0.74) remained stronger at year one and two. Compared to baseline, BMI was significantly lower at the three subsequent assessments. However, the weight loss achieved during the first year was only partly maintained at year two (Table 2). The mean weight loss from baseline constituted 4.8% after six weeks, 6.4% at year one, and 3.3% at year two.

Table 2 Mixed-effect model estimates: psychological factors, PA, and BMI during the two-year lifestyle intervention for severely obese adults.

	Baseline	Week six	Year one	Year two	
	Mean (95% CI)	Mean (95% CI)	p d	Mean (95% CI)	p d	Mean (95% CI)	p d	
Psychological factors								
Self-efficacy for PA in the face of
psychological barriersa	5.1 (4.7, 5.5)	5.6 (4.1, 6.0)	.029	5.4 (5.0, 5.9)	.141	5.5 (5.0, 6.0)	.154	
Perceived behavioural control over PAb	4.8 (4.5, 5.1)	5.4 (5.0, 5.7)	.003	5.3 (4.9, 5.7)	.026	5.4 (4.9, 5.8)	.022	
PA identityc	2.7 (2.5, 3.0)	3.1 (2.9, 3.4)	.001	3.2 (2.9, 3.5)	<.001	3.4 (3.1, 3.7)	<.001	
Accelerometer assessed PA, counts per minute	276 (241, 311)	452 (417, 486)	<.001	327 (286, 368)	.036	290 (244, 335)	.606	
BMI, kg/m2	42.1 (40.3, 43.8)	40.1 (38.4, 41.8)	<.001	39.4 (37.6, 41.1)	<.001	40.7 (38.9, 42.5)	.001	
Notes.

a Scale 1–7; higher scores represent stronger self-efficacy for PA in the face of psychological barriers.

b Scale 1–7; higher scores represent stronger perceived behavioural control over PA.

c Scale 1–5; higher scores represent stronger PA identity.

d p-values for change from baseline.

PA Physical activity

BMI Body mass index

CI Confidence interval

Significant p-values (≤0.05) in bold.

Table 3 shows the associations between change in the psychological factors and Δ PA over the two-year intervention. Δ perceived behavioural control was the only independent variable that was significantly associated with Δ PA during the two years.

Table 3 Simple and multiple linear regression analysis with Δ counts per minute as the dependent variable.

	Crude	Adjusted*	
	Reg. coeff. (95% CI)	Stand. coeff.	p	Reg. coeff. (95% CI)	Stand. coeff.	p	
Age	−1.95 (−6.45, 2.54)	−.09	.390	−1.21 (−5.66, 3.25)	−.06	.590	
Gender (refer to women)	34.73 (−66.26, 135.72)	.08	.496	13.93 (−83.85, 111.70)	.03	.777	
Δ BMI	−44.63 (−65.53, −23.74)	−.44	<.001	−39.08 (−61.81, −16.36)	−.38	.001	
Δ self-efficacy for PA	28.29 (−12.56, 69.15)	.16	.172	21.84 (−16.48, 60.17)	.13	.259	
Δ perceived behavioural control over PA	66.51 (31.40, 101.63)	.41	<.001	51.11 (16.17, 86.06)	.32	.005	
Δ PA identity	40.78 (−25.11, 106.68)	.14	.221	−20.14 (−85.07, 44.80)	−.07	.538	
Notes.

* Number of observations: 71.

Δ Change

Reg. coeff. Regression coefficients

CI Confidence interval

Stand. coeff. Standardized coefficients

BMI Body mass index

PA Physical activity

Age, gender, Δ BMI were included as covariates in the adjusted model.

Significant p-values in bold.

Discussion

In the present two-year study of associations between change in psychological factors for PA and Δ PA in severely obese adults, we found that Δ perceived behavioural control was associated with Δ PA. By contrast, Δ self-efficacy and Δ self-identity showed no association with Δ PA. Although not directly comparable, our findings differ from a cross-sectional study using the same measures which revealed positive relationships between PA and all the three psychological factors, with self-identity for PA showing the strongest association (Hansen et al., 2014).

It has been proposed that scales on perceived behavioural control reveal aspects of two different dimensions, namely control and difficulty (Sparks, Guthrie & Shepherd, 1997). With respect to the instrument used in this study, the positively worded items may capture control while the negative tap into difficulties, which could explain the Cronbach alpha of 0.67. Still, perceived behavioural control over PA was the only independent variable that worked as hypothesised. Not only was it strengthened during the intervention with a moderate ES (Ellis, 2011), but the change of it was also associated with Δ PA. Perceived behavioural control has shown cross-sectional associations with self-reported PA in adult obesity surgery patients (Hunt & Gross, 2009) and overweight and obese adolescents (Plotnikoff et al., 2013). However, to our knowledge, no studies have examined this variable during lifestyle interventions and related it to Δ PA.

Although self-efficacy, as such, is a global concept (Bandura, 1997) the measure used in this study was limited to self-efficacy in the face of psychological barriers to PA. The initial strengthening had disappeared at later assessments and was not associated with behaviour change. This could be interpreted as if the intervention did not target or succeed in strengthening self-efficacy in the face of psychological barriers, or it may indicate that psychological barriers did not play a central role in the PA of these subjects. Other barriers, such as time limitations, which we have not investigated, may be of greater significance (Biddle & Fox, 1998). Still, a longitudinal study found a positive relationship between moods and PA in overweight to obese adults with diabetes. However, the data were reported by lifestyle coaches, not patients (Venditti et al., 2014), implying a possible responder bias (Ahmed et al., 2012).

Next, the intervention strengthened the PA self-identity with a moderate ES (Ellis, 2011). Embarrassment, poor experience, and non-identification with PA may be obstacles to PA in obese adults (Biddle & Fox, 1998; Hills & Byrne, 2006). So the strengthening of PA identity could be regarded as positive (Biddle & Fox, 1998). However, in our study we could not confirm that strengthened identity translates into more PA. In community samples, PA identity has shown positive correlations with objectively measured (Hansen et al., 2014) and self-reported (Jackson, Smith & Conner, 2003) PA. Thus, this phenomenon deserves attention in future research and in clinical practice.

Regarding the impact of body weight, cross-sectional data have demonstrated an adverse relationship between BMI and objectively assessed PA (Hansen et al., 2013) and BMI and perceived behavioural control over PA (Caperchione et al., 2008). However, when controlling for Δ BMI, Δ perceived behavioural control and Δ PA still showed associations in the present study.

Overall, the findings suggest that factors associated with PA in community samples (Jackson, Smith & Conner, 2003; Lorentzen, Ommundsen & Holme, 2007; Hansen et al., 2014) should not be generalised to samples of severely obese adults in lifestyle interventions without caution and testing. Social and environmental factors, including family, work place, and community, may predict and mediate the mechanisms of change in PA in this population (Vartanian & Shaprow, 2008; Wiklund, Olsén & Willén, 2011). Thus, future research could take broader perspectives and adopt an ecological approach (Bauman et al., 2012).

Our study confirms the findings from other studies (Borg et al., 2002; Tate et al., 2007) that maintenance of PA is an unresolved challenge. With regards to the overall PA, the initial and year two PA (Table 2) were similar to the PA of American obese adults (288 CPM) (Tudor-Locke et al., 2010) and their Norwegian counterparts (women: 276 CPM, men: 290 CPM) (Hansen et al., 2013), whereas the mean value from the second assessment (Table 2) was well above the 344 CPM for American normal weight (Tudor-Locke et al., 2010) and 352 CPM for women and 368 CPM for men of normal weight in Norway (Hansen et al., 2013).

Regarding weight loss, the one-year reduction of BMI (Table 2) was within the criterion for success, defined as 5–10% reduction from the start of an intervention (Tsigos et al., 2008; Dalle Grave, Calugi & El Ghoch, 2013). However, patients had regained some of the weight at year two which is a common challenge in lifestyle interventions (Dalle Grave, Calugi & El Ghoch, 2013).

The present study offered novelty and strength as it used data from four time points and therefore could provide information about patterns of change throughout the two-year intervention. In addition, assessing PA objectively with accelerometers is superior to self-reported PA (Prince et al., 2008). However, accelerometers fail to capture water activities, bicycling, and strength training (Warren et al., 2010). For the present study, this limitation probably caused a 25% underestimation of the true overall PA for the second assessment (Aadland et al., 2014), as such activities were common during the residential period. Still, for the purpose of the study, we decided to avoid reporting of intensity-specific PA, due to difficulties of interpretation when applying count thresholds to separate different intensities of PA generally (Orme et al., 2014) and in the severely obese population specifically (Aadland & Steene-Johannessen, 2012). Underestimation of PA was probably a minor problem when assessing trends over the home periods, because patients generally did not engage in such activities (Aadland & Robertson, 2012), and because the underestimation would be equally distributed over time.

The main weakness of this study was the relatively high proportion of drop-outs and missing data. Although the drop-out analysis did not reveal differences between the completers and the non-completers, bias cannot be ruled out. However, by using the mixed model based on maximum likelihood estimation and including all valid observations from all four time points, the statistical power increased. Still, our results are based on associations and thus, causal relationships cannot be inferred. Lastly, the participants were a self-selected, treatment-seeking group, participating in a specific intervention programme and there was no control group. While common in clinical studies, these weaknesses limit the generalisability of our results. For transparency and usefulness, we have therefore attempted to report rigorously on the intervention and the flow of the participants (Vandenbroucke et al., 2007).

Conclusion

Little is known about factors related to the process of change of PA behaviour in severely obese adults participating in lifestyle interventions. We hypothesised that the independent variables: Δ self-efficacy for PA, Δ perceived behavioural control over PA, and Δ PA self-identity would be associated with the dependent variable of Δ PA in the sample of severely obese adults who participated in a two-year programme. However, such an association was only confirmed between Δ perceived behavioural control and Δ PA. More research is required to investigate PA behaviour change processes in severely obese both in non-residential and residential settings and with larger samples and stronger design. An ecological framework may provide a good structure (Bauman et al., 2012), with both quantitative and qualitative methods being suitable.

The findings of the present study indicate that perceived behavioural control may be a valid target for increase and maintenance of PA in severely obese adults.

We thank the staff at RCHRC for their assistance in the data collection and express our sincere gratitude to the participants.

Additional Information and Declarations

Competing Interests

Author Contributions

Human Ethics

Lesley Robertson is employed by the Red Cross Haugland Rehabilitation Centre and John Roger Andersen is employed by the Førde Health Trust. The authors declare there are no competing interests.

Randi Jepsen conceived and designed the experiments, performed the experiments, analyzed the data, wrote the paper, prepared figures and/or tables, reviewed drafts of the paper.

Eivind Aadland conceived and designed the experiments, performed the experiments, analyzed the data, reviewed drafts of the paper.

Lesley Robertson reviewed drafts of the paper.

Merete Kristiansen analyzed the data, reviewed drafts of the paper.

John Roger Andersen and Gerd Karin Natvig conceived and designed the experiments, analyzed the data, reviewed drafts of the paper.

The following information was supplied relating to ethical approvals (i.e., approving body and any reference numbers):

Ethical approval was obtained from the Regional Committee for Medical and Health Research Ethics for South-East Norway (registration number 2010/159).

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
