# Peer review of "Factors and associations for physical activity in severely obese adults during a two-year lifestyle intervention"

_PeerJ, doi:10.7717/peerj.505_

## Round 0.1 · original submission · Major Revisions

There are concerns about the small sample size at year 2 due to attrition. Would it be possible to report the data up to year 1 to maximize the sample? This would, of course, require major revision to the abstract, results, discussion and conclusion.

Please describe the changes to the intervention that were created by reduced financing. In addition, please describe the problems with the study protocol that led to 5 participants' withdrawal..

Reviewer 1 ·

Basic reporting

Introduction:
• Line 44: Define “PA capacity”
• Line 53: Give examples of psychological barriers
• Lines 60-62: Provide similarly matched definition of PA self-identity (e.g. “belief that…..”

Experimental design

Materials and methods:
• Line 94: Include comorbidities included
• Line 101: Did the financing and change in intervention influence retention as well? If so, this should be discussed further in the limitations.
• Did the intervention include specific dietary, physical activity, or weight loss goals? If so, did they differ between residential and home?
• Line 117: Did all participants engage in 11-12 hours/week of physical activity? This is an extraordinary amount of physical activity. How were they transitioned from residential (11-12 hours/week of physical activity) to home (~30 minutes/day)?
• Line 137: It would be interesting to see how many physical activity diaries were returned. Was this information tracked?
• Lines 150-166 – It would be easier to follow if references were following each description of the questionnaires, rather than all at the end. Also, please clarify how the scales were moderated (line 161).
• Line 178 and limitation section lines 326-327 – It is unclear why you would chose to use the average summed total of counts rather than using intensity cut points to determine number of minutes spent in MVPA. I have never seen this method done before. Furthermore, knowing that overall intensity increased from 276 to 452 counts/minute is not meaningful, particularly as moderate intensity starts at least at 1,900 counts/minute. While there will always be the limitation of using an accelerometer in severely obese adults, having minutes of physical activity would be more helpful. It would also be more comparable to other research as well as the physical activity guidelines to increase physical activity to 150 min/week or 300 min/week (weight loss).

Validity of the findings

Results:
• Line 238: Why were participants dropped if they met their study goal? Were they no longer eligible to participate in the residential period?
• Lines 239-240: Having 10% of the sample drop out because of the “study protocol” seems unusually. Please clarify.
• Reporting changes in weight and weight loss percentage would be helpful
Discussion:
• Lines 326 – As mentioned above, this manuscript would be stronger if it reported time spent in physical activity (MVPA or even light intensity physical activity), rather than an average summed total of counts.
• Lines 332 – The limitation section should further address the poor retention. Given the high attrition, it is hard to make any definite conclusions.
• Lines 310-316: The sample included in this study is much more specific than a community sample, as well as general adults who participate in lifestyle interventions. A lifestyle intervention that incorporates residential periods has a sample that is unique and not generalizable.

Conclusion:
• While impressed by the openness in which you laid out the study’s limitations, the conclusion was lacking a more persuasive reasoning of why the study was important. Left the reader questioning the studies validity, reliability, and generalization. Need to state how this will help healthcare community settings.

Additional comments

This study examine the association between objectively assessed physical activity and change in perceived self-efficacy, behavioral control, and self-identity during a two-year multiple health behavior intervention for severely obese.
Strengths: The study had a strong introduction, which well outlined the issue at hand and the current literature . As a whole, manuscript was well-written and organized.
Weaknesses: Attrition and the way physical activity was calculated are the major weaknesses of this manuscript. In general, some components lacked through explanation and rational for study weaknesses need be explained in more detail. Given the high attrition, the results add little to the literature.

Reviewer 2 ·

Basic reporting

no comments

Experimental design

no comments

Validity of the findings

I am concerned by the high drop out rate and small sample size, questioning whether there was enough power to examine this research question.

I am also concerned that the PA monitor likely didn't pick up 25% of the PA being performed by the subjects at the 6 week time point, since water activities, cycling, and weight training were prevalent. Thus, how can you look at the relationship between PA and psych outcomes if you don't have a good measure of PA?

Additional comments

This study assessed the relationship between changes in self-efficacy, perceived behavioral control, and self identify with objectively measured physical activity among severely obese adults participating in a two-year lifestyle intervention. While this study has the potential to improve our understanding of PA behaviors in a severely obese cohort, I have several concerns including the high drop-out rate, small sample size, and the examination of physical activity in counts/min which isn't clinically relevant to the reader. See below for major/minor comments.

Major concerns:
1. The drop out rates in this study were extremely high (~50% at year 2), thus reducing an already small sample size (n=53 ppts initially) to n=22 with complete data at year 2. I question whether this sample size is sufficient to answer the proposed research question. In addition, we know from previous research that individuals who drop out of programs are often those who were disengaged or those who did poorly in the program. Therefore, the findings presented may not be reflective of the entire starting cohort and is a significant limitation of this paper. Have you considered only examining this research question up until the 1-year time point to maximize your sample?

2. The objective PA data is presented in counts/min. While this may (or may not) be an appropriate method to examine the relationship between total PA and psychological outcomes, it has no clinical meaning to the reader. For example, I do not know whether a change in PA from baseline (276 counts/min) to week 6 (452 counts/min) is even clinically meaningful. I would suggest using a standardized cut-points for the Actigraph to at least present the amount of time spent in moderate-to-vigorous intensity PA at each time point within Table 1. Also, as written it is difficult to know whether the variation at each time point is typical or not. Physical activity in obese cohorts is notorious for very high levels of variation between individuals. Is it possible that the lack of correspondence between the psychological variables and PA observed could be partly related to the variability in PA observed within this sample? Finally, by simply looking at the association between the psych variables with PA expressed as counts/min, isn't it possible that participants increased their "structured" PA (is this what was recommended in the intervention?), but then compensated by being less active throughout the day? If this were the case, there would be no difference in counts/min, but there would likely be a difference in mod to vig PA if cut-points were used. I understand that there is debate in the field over the use of cut-points, but at the very least I suggest including both counts/min and minutes spent in mod to vig activity.

3. In the Discussion the authors state that accelerometers fail to capture water activity, bicycling and weight training and "for the present study, this limitation probably caused a 25% underestimation of the true PA level at the second assessment as such activities were common during the residential stay." I have two concerns with this. 1) was the 6-week PA measurement taken WHILE ppts were still at the clinic? If so, this would skew the results because it appears that 1 and 2-year PA data was collected either before or after the clinic visits. 2) A 25% underestimation of PA is a lot and is a significant limitation of this paper. If ppts were instructed to swim, cycle and weight train (as indicated in the Methods), and accelerometers do not pick up these data, how can you assess the relationship between PA and psych variables when your PA measure isn't capturing all (maybe not even most) of the PA that is being performed. While the use of an objective PA monitor is usually advantageous compared to self-report, that may not be the case in this instance, given that ppts were performing a lot of activity that is not captured by PA monitors.

Minor concerns:
- Given that this was a lifestyle intervention, I believe that it is equally important to also report changes in weight (not just PA) throughout the intervention period (this can be added to Table 1). BMI changes are reported in the text, but kg or % WL would be a nice addition to Table 1. Changes in physical activity self-efficacy, behavioral control, and self-identify could also be tightly linked with changes in weight (e.g., someone could lose a lot of weight without increasing PA and still experience changes in these psychological variables, simply as a byproduct of either the weight loss itself or feeling more in control of their eating habits, which could translate to their views on PA as well). Therefore, this is a concern with the regression analyses presented in Table 2. Within the Discussion, the authors state that the inclusion of change in BMI did not change the results of the regression analyses and mention that the data is not shown, however, I suggest adding these data to the results. Were there associations between changes in weight and changes in these psych variables?

- In Table 2, it is unclear whether this is simply the change from baseline to 2-year or whether this includes the intermediate times points as well. The text indicates that this may be the entire 2-year period. Please clarify. Also, the footnote of this table should indicate which variables were included in the "adjusted" model.

- What was the exercise prescription given to the participants for when they were at home, and not staying at the residential clinic? Do you have any indication of whether individuals were doing the amount of activity that was prescribed?

- Overall, the paper seemed rather wordy and lengthy and could be tightened up, particularly within the Introduction, Intervention, and Statistical Analyses sections.

Reviewer 3 ·

Basic reporting

It would be helpful if the authors can provide more background on the effectiveness of the weight loss strategies described in the introduction (lines 26-30). Particularly, emphasis should be placed on the interventions targeting the severely obese population.

Lines 41-43: Are the barriers severely obese individuals face significantly different compared to other populations? Also, there is evidence to suggest that increasing PA in this population is feasible (Goodpaster et al., Effects of diet and physical activity interventions on weight loss and cardiometabolic risk factors in severely obese adults: a randomized trial).

Line 53, 54: A psychological barrier is a term used but it is never specified what barriers these refer to (e.g., self-confidence, depression, etc…)

Line 55: Does community samples relate to lifestyle interventions? The first paragraph of the introduction suggests that you are examining the effect of a lifestyle intervention on psychological factors which may influence physical activity.

Experimental design

Lines 116-117: It appears the patients completed a high volume of PA during the residential periods (11-12 hrs/wk). This may influence perceived behavioral control over PA.

Lines 118-119: Please specify if patients were prescribed physical activity goals for at home.

Lines 130-131: Please report how many patients needed individual CBT and attendance rates at in-house sessions.

Lines 136-137: Please clarify if patients self-monitored diet/physical activity behaviors or only physical activity. Also, please include the percentage of patients who completed diaries. Was this associated with greater PA levels?

Lines 165-166: Positive associations between PA and psychological factors are observed in adults? It is unclear what this statement is referring to.

Lines 178-179: It would be interesting to examine if psychological factors varied by intensity levels (sedentary, light, and moderate-vigorous PA) in comparison to counts/min.

Lines 183, 199-203: Sociodemographic data in table would be helpful for reader (including change values for four time points).

Validity of the findings

In order for the data to be robust, baseline data/last observation should be carried forward.

Line 219: Was weight change a covariate?

Line 238: Why would reaching weight goal be a reason for drop out?

Lines 257-262: The weight change was modest for an intervention which appeared to focus on diet and physical activity.

Line 271: The p-value for change from baseline PA to Year 2 indicates there was no difference.

Lines 326-328: It make the findings more applicable to the reader, I would recommend reporting results in minutes/week if intensity levels are not able to be reported. This would allow the results to be more robust.

Lines 351-352: The findings suggest that perceived behavioral control may assist with the maintenance of physical activity, not the increase as there was no change in PA from baseline to 2 years.

---

## Round 0.2 · accepted · Accept

Thank you for your recent edits to the manuscript. They provide more clarity.